# Prevalence of Drug and Substance Use among Malaysian Youth: A Nationwide Survey

**DOI:** 10.3390/ijerph19084684

**Published:** 2022-04-13

**Authors:** Rozmi Ismail, Mohd Rizal Abdul Manaf, Mohd Rohaizat Hassan, Azmawati Mohammed Nawi, Norhayati Ibrahim, Novel Lyndon, Noh Amit, Ezarina Zakaria, Muhammad Ajib Abd Razak, Norshaffika Izzaty Zaiedy Nor, Md Shafiin Shukor, Aimi Fadzirul Kamarubahrin

**Affiliations:** 1Psychology and Human Wellbeing Research Centre, Faculty of Social Sciences and Humanities, Universiti Kebangsaan Malaysia, Bangi 43600, Selangor, Malaysia; rozmi@ukm.edu.my (R.I.); ezaz@ukm.edu.my (E.Z.); muhdajib@ukm.edu.my (M.A.A.R.); shaffika.izzaty@ukm.edu.my (N.I.Z.N.); 2Department of Community Health, Faculty of Medicine, Universiti Kebangsaan Malaysia, Jalan Yaacob Latiff, Bandar Tun Razak, Cheras, Kuala Lumpur 56000, Selangor, Malaysia; rohaizat@ppukm.ukm.edu.my (M.R.H.); azmawati@ppukm.ukm.edu.my (A.M.N.); 3Health Psychology Programme, Faculty of Health Sciences, Universiti Kebangsaan Malaysia Kuala Lumpur Campus, Kuala Lumpur 50300, Selangor, Malaysia; yatieibra@ukm.edu.my (N.I.); nohamit@ukm.edu.my (N.A.); 4School of Development, Social and Environmental, Faculty of Social Sciences and Humanities, Universiti Kebangsaan Malaysia, Bangi 43600, Selangor, Malaysia; novel@ukm.edu.my; 5UKM Pakarunding, Universiti Kebangsaan Malaysia, Bangi 43600, Selangor, Malaysia; p80550@siswa.ukm.edu.my (M.S.S.); aimifadzirul4@gmail.com (A.F.K.)

**Keywords:** prevalence, substance and drug abuse, youth, Malaysia

## Abstract

Uncontrolled drug and substance use (DSU) may pose unprecedented threats to nation building and a country’s socioeconomic development. Despite considerable efforts made and resources used to address DSU concerns, Malaysia has seen a significant annual rise in cases of DSU. The bulk of the reported cases originate from youth between the ages of 15 and 40 years. To date, data related to DSU in Malaysia have been entirely dependent on operation statistics, arrest counts, and reported cases; DSU may therefore be under-reported and the data obtained not representative at the national level. This study aims to determine the prevalence of DSU among Malaysian youth through a large nationwide representative survey. Of the population of 11,129,316 youth aged 15–40 years, the prevalence of DSU among lifetime users was found to be 5.5%, while for those who had taken drugs in the past 30 days or who currently use them, the prevalence of DSU was found to be 3.5%. The most popular drugs for lifetime users were kratom or *Mitragyna speciosa*, while for current users the most popular drug was cannabis. The current study reports the magnitude of the problem at a country-wide level, which is a crucial preliminary effort for crafting evidence-based and well-informed policies.

## 1. Introduction

Drug and substance use (DSU) is a worldwide problem [1,2]. The abuse of psychoactive substances is associated with many harmful effects on both physical and mental health. It not only causes various social problems such as violence and crime, but also has a tendency to catalyze serious mental disorders and lead to greater susceptibility to HIV/AIDS, Hepatitis B or C, and tuberculosis infections [3,4,5]. DSU has a major impact on health care services, public services, and the criminal justice system [6]. The bulk of the health care budget is spent on treating the effects of addictive drugs [7]. Every year, the Malaysian government spends over half a billion Malaysian Ringgit to deal with drug offenders, such as by arresting and housing addicts in state-run rehabilitation centers throughout the country [8]. Delayed interventions to curb DSU would have detrimental effects on the country’s public health system, causing political insecurity as well as economic and societal collapse [9,10,11].

According to Trendwise Analytics, DSU has accelerated addiction among youth, including high school and university students, causing long-term impacts on individuals’ socio-behavioral characteristics [12]. Chronic addictions have the tendency to cause a spectrum of behavioral disorders, from social delinquency such as disciplinary problems in school (skipping and dropping out of school) to mental health repercussions such as depression or violent or criminal behaviors [4,5,13].

According to the World Drug Report [9], an estimated 13.8 million people aged between 15 and 16 years (5.6%) are users of cannabis-type drugs. In Malaysia, approximately 2138 adolescents or 9.25% abused drugs in 2010 [14], with a majority of them being youths aged 19–39 years, followed by those aged 13–18 years [14]. According to the Institute for Public Health report conducted by Ministry of Health Malaysia (IPH) in 2019, the prevalence rates of DSU within the past 30 days and of drug use among Malaysians aged 18 years and above were 0.5% and 1.5%, respectively. The report highlighted that the prevalence of kratom use (a herbal substance that can produce opioid- and stimulant-like effects) was 0.4% within the past 30 days, and that of cannabis use was 0.7% for one-time drug users [15].

Other countries such as India are also facing similar problems, as a majority of drug users are aged between 13 and 15 years and have no permanent income or job. The prevalence of DSU reported in an Indian study was 66%, with 48.54% of the subjects having ever used of alcohol and 23.36% having consumed tobacco [16]. A study by de la Torre-Luque, Ozeylem, and Essau [17] on addictive behaviors among adolescents from 73 low- and middle-income households found that the prevalence of substance use in Asian countries was high among adolescents, especially in non-Muslim countries such as Laos, the Philippines, and Thailand compared with countries with majority-Muslim populations (Senegal, Indonesia, and Malaysia). A study by Hong and Peltzer [18] also reported that the prevalence of current substance use among adolescents in Association of Southeast Asian Nations (ASEAN) countries was between 11% and 15%.

Adolescents are the group most prone to addiction [19]. They have a strong inclination toward experimentation, curiosity, susceptibility to peer pressure, and poor self-worth, which makes them vulnerable to drug abuse [10]. The initiation of drug use generally begins during adolescence, and the maximum usage of drugs occurs among youth aged 18–25 years [9]. Most studies have shown that a person who takes a drug at a young age has a high risk of becoming addicted [20] and an increased risk of substance abuse problems in the future [21]. Adolescents who began consuming alcohol before the age of 14 were found to have a 40% higher risk of experiencing substance abuse problems when entering early adulthood [22,23]. In addition, substances that were previously not considered illegal drugs have now become abused substances [24]. This has led to the misuse of drugs such as marijuana or kratoms and raised concerns about addiction to more serious drugs such as cocaine or heroin. Two crucial aspects for policy-related developments in curbing drug misuse are data accuracy and availability. 

Unfortunately, no studies have yet been conducted on the prevalence of DSU among Malaysian youths. The available studies, conducted by stakeholders such as the Malaysian Ministry of Health, have focused on epidemiological issues rather than on drug addiction. The results do not reflect the real situation of DSU at the present time. In-depth research is therefore needed to determine the prevalence of drug involvement, especially among youths, in terms of the onset stage [25]. This research is important for reporting the latest data on the prevalence of DSU in Malaysia. Malaysia has lagged behind other ASEAN countries such as Thailand, which is actively conducting a ‘National Survey on Drug Use’. Malaysia only estimates the number of its drug addicts based on the ratio released by the World Drug Report (WDR), which is 1:8 (for every addict arrested, there are eight other addicts still undetected) [1].

## 2. Materials and Methods

### 2.1. Study Setting

This nationwide interviewer-administered cross-sectional survey was conducted from January 2020 to July 2021 throughout Malaysia. Samples were randomly selected using enumeration blocks (EB) and living quarters (LQ), which were provided by the Department of Statistics Malaysia (DOSM). 

All respondents from a selected household who met the following two criteria were included in this study: (a) aged 15–40 years (from the definition of youth based on the Malaysian Youth Council and the Ministry of Youth and Sports, Malaysia, as stated in the Youth Societies and Youth Development Act 2007) and (b) without any acute psychiatric illness. Non-Malaysians were excluded from the study. This study received ethical approval from the Universiti Kebangsaan Malaysia Research Ethics Committee (UKM PPI/111/8/JEP-2020-583). Written consent was obtained from each eligible respondent prior to enrolment.

### 2.2. Sample Size and Sampling Strategy

The required sample size was calculated based on the estimated prevalence rate of 5% of current users reported by the Institute for Public Health Malaysia [15]. The accuracy was 0.95 and the level of significance was set at 0.05, with a Z statistic of 1.96 and a precision of 7%, thus yielding a minimum sample size of 3724. Based on complex sampling design, a sample size accounting for a design effect of 2 yielded a minimum sample size of 7448. Furthermore, accounting for a non-response rate of 30% [26], the total sample size for this study was 10,640 youths. For each state, using the random sampling method, a total of *n* = 391 was required. However, to avoid over-sampling for states with lower populations or under-sampling for highly populated states, the probability-proportionate-to-size sampling (PPS) technique was used to estimate the number of youths in each state to whom the study had access through the DOSM statistics. This means that the total sample for the general youth population throughout Malaysia was 10,640. 

However, during the data purification process, 7% of the participants were removed because they were recruited from outside the enumeration blocks (EB) list. The final total number of respondents for this study included in the analyses was 9818 (Figure 1). This sample number did not affect the research requirement because the sample calculation considered the dropout rate to be 30%. 

### 2.3. Data Collection and Research Instruments

Since the data collection method used was face-to-face, each enumerator was briefed on the Universiti Kebangsaan Malaysia (UKM) COVID-19 fieldwork safety protocol. Trained enumerators administered validated instruments during the house-to-house data collection and were supervised by field managers. The components of the survey were as follows. Part A comprised the socio-demographic domain, which captured information on respondent characteristics, such as gender, age, personal income, and household income. In Part B, the Alcohol, Smoking, and Substance Involvement Screening Test (ASSIST) scale [27] was utilized to measure the frequency of substance and drug intake by type. This scale was designed and developed by an international group of researchers for the World Health Organization (WHO) to detect and control the symptoms of substance and drug use. In other words, ASSIST aims to identify any problems related to substance and drug use within the last 3 months quickly and effectively [27]. ASSIST determines a risk score for each substance by placing it in a lower-risk, moderate-risk, or high-risk category. It obtains information from respondents about their lifetime use of substances, as well as problems associated with substance use (acute intoxication, regular use, dependent and injecting behavior) over the last three months (current users). The substances studied in this test were cigarettes, alcohol, marijuana, stimulants, sedatives, and opiate drugs, as well as any other substances filled in by the participants. The data were collected electronically by the enumerators using a software application. The verification of the data was conducted by field managers and the researchers.

### 2.4. Screening Process and Determination of Severity Rate of Drug Use 

To simplify the description of DSU prevalence in Malaysia, the drug use screening process consisted of two categories—namely, lifetime users (users who at some point in their lives have used the drug before the time of assessment) and current users (users of the drug within the past 30 days). Next, the category of current users was subdivided into three levels of severity based on ASSIST scores—namely, mild (0–3), moderate (4–26), and severe (27 and above) [27].

### 2.5. Statistical Analyses

Analyses of the data were performed based on complex samples analysis (CSA) using the Statistical Package for the Social Sciences (SPSS) version 26 for data analyses (IBM Corp., Armonk, NY, USA). The CSA takes into account the weighting factor (W) of each respondent in order to adjust for non-response and the probability of having a different number of samples needing to be adjusted according to the population. The weights (W) used were as follows:W=W1× F × PS
where:
W1 is the inverse of the probability of selecting the EBs;F is the non-response adjustment factor;PS is a post-stratification adjustment factor calculated by state, rural or urban status, age, gender, and race.

In SPSS, descriptive analyses were performed to describe the characteristics of the participants, in which frequencies and percentages for demographic categories and prevalence were presented [28]. A 95% confidence interval (CI) was used for the population parameter in the prevalence analysis.

## 3. Results

### 3.1. Participant Characteristics

A total of 9818 participants were included for the final analysis, since 7% of the informants were removed and replaced by respondents from outside the enumeration blocks (EB) list. Table 1 summarizes the participants’ characteristics. The majority of the youths were males (55.7%) and aged between 31–40 years (36.6%). Most youths were single (51.8%), Malay (64.9%), and living in village-type houses (36.9%) within neighborhood settlements (57.4%). These participants had been living in their current neighborhood for about 6–10 years (26.6%). Most of them had completed secondary education (49.2%), and 66.0% were in the B40 household income group (bottom 40% of household income classification in Malaysia).

### 3.2. Prevalence of Substance and Drug Use

Table 2 shows the prevalence of substance and drug use. The overall prevalence of the lifetime use of tobacco, alcohol, and drugs was 22.4%, with respondents aged between 31 and 40 years reporting the highest prevalence of drug use (26.2%). The overall prevalence of current use of tobacco, alcohol, and drugs was 12.9%, with respondents aged 31–40 years reporting the highest prevalence (15.8%).

### 3.3. Prevalence of Substance Abuse by Type According to Age Categories

Table 3 shows the prevalence of tobacco and alcohol consumption by age category. The analysis found that the overall prevalence of lifetime tobacco use was 19.9%, while that for lifetime alcohol consumption was 4.9%. The prevalence rates for lifetime tobacco use and alcohol consumption were the highest among respondents aged 25 to 30 years: 22.8% and 6.1%, respectively. More concerning is the prevalence of current tobacco and alcohol use among those of secondary school age (15 to 18 years old), which was found to be 6.6% and 2.4%, respectively. 

Table 4 shows that the overall prevalence rate of current tobacco use was 12.5%, while that of current alcohol consumption was reported to be 3.5%. The prevalence rate for current tobacco use was highest amongst respondents aged 31–40 years (13.7%). By contrast, the prevalence rate for current alcohol consumption was highest among respondents aged 31–40 years, accounting for approximately 4.5% of the total sample.

### 3.4. Prevalence of Lifetime and Current Drug Use by Age Group

Table 5 shows the prevalence of lifetime and current drug use by age group. The overall prevalence of lifetime drug users was 5.5%, while for current use it was 3.5%. The prevalence of lifetime and current drug users was highest in the group aged 25–30 years, with 7.7% and 4.8%, respectively, being reported.

### 3.5. Level of Severity for Current Drug Use 

Current drug use was classified according to three levels of severity—mild (0.3%), moderate (2.3%), and severe (1.0%)—for the population of 11,129,316 youth. Table 6 compares the prevalence of current drug use by severity according to rising age groups. While the mild level of current drug use showed an S-shaped pattern with rising age, moderate and severe drug use showed an upward trend until peaking at 25–30 years of age and subsequently down-trending to 31–40 years of age. 

### 3.6. Prevalence of Drug Use Based on Demographics

Table 7 exhibits the prevalence of drug use based on demographic profiles. Significant differences in the prevalence of lifetime and current drug use were observed in males (10.1% vs. 46.6%), singles (6.1% vs. 4.3%), those with a secondary education (6.2% vs. 4.3%), and those in the B40 household income group (7.9% vs. 5.1%). 

### 3.7. Prevalence of Drug Use by Drug Type

Table 8 shows the prevalence of drug use by drug type. The highest prevalence rate for drug types used for lifetime use was for kratom or *Mitragyna Speciosa* (2.8%), followed by cannabis (2.5%) and amphetamine-type stimulants (ATS) (2.0%). The highest prevalence rate by drug type for current drug use was for cannabis (1.9%), followed by ATS (1.7%) and kratom (1.5%).

### 3.8. Prevalence of Polydrug and Non-Polydrug Use

Polydrug use is a term for the use of more than one drug or type of drug at the same time or one after another [29]. Polydrug use can involve both illicit drugs and legal substances, such as alcohol and medications. Generally, the purpose of polydrug use is to enhance the desired effects of one drug, such as drinking alcohol while using stimulants or substituting a drug of choice for an alternative—for example, being unable to access heroin and substituting alcohol and cannabis for it instead [30]. The prevalence of lifetime non-polydrug and polydrug use was 4.1% and 1.4%, respectively. The prevalence of non-polydrug and polydrug use amongst current users was 2.3% and 1.2%, respectively.

## 4. Discussion

### 4.1. Summary of Core Findings and Comparisons with Existing Literature

This study aimed to determine the prevalence of DSU among youths aged 15 to 40 years in Malaysia. The results found that a total of 22.4% of the participants admitted to taking drugs and substances (alcohol, tobacco, and cigarettes) in their lifetime and more than half (12.9%) reported being current users. Nearly one-fifth (19.9%) had used tobacco products (cigarettes, chewing tobacco, cigars, and others) in their lifetime and almost half as many (11.5%) were current tobacco product users. Moreover, 4.9% had consumed alcoholic beverages such as beer, wine, liquor, and others in their lifetime and only 3.5% were current drinkers. By contrast, 5.5% had used drugs in their lifetime and 3.5% were current users. The prevalences found were higher than those previously reported in Malaysia based on the National Health and Morbidity Survey (NHMS) among adolescents aged 13 to 17 years in 2017 and among adults (18 years and above) in 2019 [15,30]. The NHMS report stated that the prevalence of current cigarette smokers was 13.8%, compared with 10.2% for consumers of alcohol; furthermore, 4.3% reported being lifetime drug users and 3.4% current drug users. However, this study generally found lower prevalence rates compared to studies from abroad, such as in the United States, Norway, and Iran [31,32,33,34,35,36]. These differences in findings may be due to this study’s method of data gathering (i.e., face-to-face interviews), sampling size, and respondent characteristics (youth aged 15–40 years), since other studies only used high school and university students as their samples. Other factors may be due to differences in culture [37] and social ceremonies [38,39]. In addition, inherited ritual ceremonies may sometimes lead to substance intake that corresponds to social norms [40,41] or to beliefs such as those reflected in orally transmitted traditions related to the substance types [42,43,44]. Conversely, drug and alcohol consumption may also be incompatible with local norms and culture, especially for Muslims. Furthermore, the respondents’ geographical location may also have affected the findings, especially for those who live near the country’s international border entrances [45,46]. However, a study conducted by de la Torre-Luque and colleagues [17] found that tobacco was the substance most frequently used among adolescents in Indonesia, Laos, Malaysia, Myanmar, the Philippines, and Thailand.

The study findings also revealed that the prevalence of lifetime drug users among Malaysian youth was 5.5%, while 3.5% of the total youth population had taken drugs within the past 30 days. The values obtained in this study were relatively high compared to the findings reported by the Institute for Public Health (IPH), Ministry of Health Malaysia, in 2019 [15]. The IPH research found that the prevalence rate was 1.5% for those aged 18 years and above. A discrepancy in sampling could have caused the differences in results; the number of respondents used by NHMS was 16,688, whereas this study used 9818. Nevertheless, the findings of this study are relatively similar to those of the research of Yi, Peltzer, Pengpid, and Susilowati [47], which focused on university students in several ASEAN countries, including Cambodia, Indonesia, Laos, Malaysia, Myanmar, the Philippines, Singapore, Thailand, and Vietnam, in 2015. Their study also showed a relatively high prevalence rate (16.9%). Another study by Ah Hong and Peltzer [18] also found the prevalence of substance use among Malaysian adolescents to be high for both boys and girls: 30–40% and 10–20%, respectively. 

In every category used, the results show that the drugs most often consumed by Malaysian youth are kratom, cannabis, and ATS. This contradicts the findings of the IPH [15], in which cannabis was found to be the most prevalent drug. However, in the present study, for the category of drugs used within the last 30 days, the most frequently used drugs were cannabis, ATS, and kratom. By contrast, the IPH found that the drug most frequently used within the past 30 days was kratom. 

In addition, the prevalence rate of severe addiction among Malaysian youth was 1.0% (*n* = 146, *N* = 110,337) of the population of 11,129,316 youth represented in this study. The most popular drugs among addicts were ATS, cocaine, and cannabis. This corresponds to research conducted by the United Nations Office on Drugs and Crime (UNODC) in which respondents aged between 18 and 25 were found to have the highest tendency toward drug abuse, a result which can be observed in most European countries and England [46]. The tendency toward the abuse of amphetamines and ecstasy was two to three times higher among those aged 35 years and below. Based on further UNODC research [48], youths were found to be more likely to use prohibited substances compared to other age groups. This study was conducted during the COVID-19 pandemic, and the unprecedented nature of this pandemic may have intensified pre-existing social determinants. According to the Centers for Disease Control and Prevention [49], since June 2020 there has been an increase in substance use as a way to cope with stress or emotions related to COVID-19. Overdoses have also spiked since the onset of the pandemic. 

### 4.2. Study Limitations and Recommendations

Some limitations of this study should also be considered when interpreting our findings. The survey was targeted towards youths aged 15–40 years, and since it is more likely that individuals will respond to surveys if they see topics or items that are of interest to them, those who chose to respond differ by definition from those who did not participate. Some of the selected respondents also hesitated to tell the truth about their past experiences and were unable to communicate well about what they were asked to share, especially about drug-related events and activities in the past. In addition, some respondents remained fearful and suspicious, since they were worried about how the information supplied to the researchers might be used. It is also important to be aware of the predictive limitations of cross-sectional studies, of which ours is an example: the exposure and outcome are simultaneously assessed and there is generally no evidence of a temporal relationship between exposure and outcome. Therefore, it is difficult to draw predictive conclusions based on these differences. It is recommended that an approach be used which can obtain access to sections of the populations that are more hidden and hard to reach (including adolescents attending school, college students, professionals, and blue-collar workers).

## 5. Conclusions

Overall, this study demonstrated that the current prevalence of DSU among youths in Malaysia is higher than that previously reported among adolescents and adults in Malaysia. The results also show that the drugs most frequently consumed by Malaysian youth are kratom, cannabis, and ATS. Further research is needed to confirm these findings, especially to ascertain whether the unprecedented COVID-19 pandemic influenced the pre-existing factors disposing youth towards DSU. This study also drew attention to the urgent need to strengthen existing intervention programs, drug policies, and professional supports, or to devise improved ones, with the aim of reducing DSU among youths in Malaysia.

## Figures and Tables

**Figure 1 ijerph-19-04684-f001:**
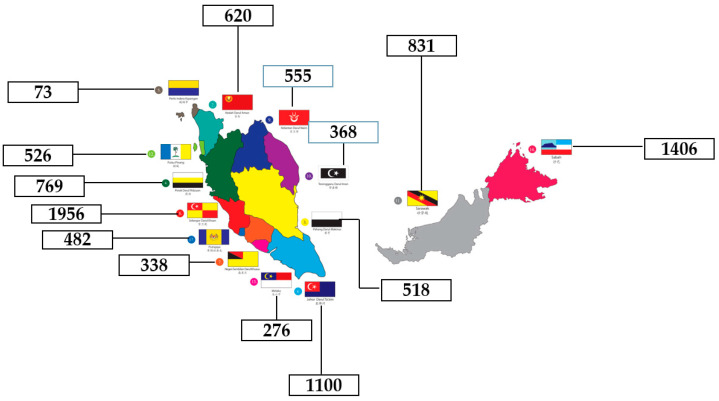
The number of samples taken according to state in Malaysia.

**Table 1 ijerph-19-04684-t001:** Malaysian youth demographic profiles (*n* = 9818).

Demographic Profiles	Frequency (*n*)	Percentage (%)
**Age:**		
15–18 years	959	9.8
19–24 years	2541	25.9
25–30 years	2729	27.8
31–40 years	3589	36.6
**Gender:**		
Male	5464	55.7
Female	4354	44.3
**Marital Status:**		
Single	5086	51.8
Married	4252	43.3
Divorced	480	4.9
**Race:**		
Malay	6369	64.9
Chinese	971	9.9
Indian	867	8.8
Bumiputera Sarawak	604	6.2
Bumiputera Sabah	957	9.7
Others *	50	0.5
**Religion:**		
Muslim	7765	79.1
Buddhist	644	6.6
Hindu	868	8.8
Christian	515	5.2
Others **	26	0.3
**Type of House:**		
Bungalow	474	4.8
Semi-Detached	434	4.4
Terrace	3212	32.7
Flats	990	10.1
Apartment	696	7.1
Condominium	32	0.3
Traditional Village	3610	36.8
Long House	370	3.8
**Type of Settlers:**		
Traditional Village	3626	36.9
FELDA/FELCRA Settlers	72	0.7
Estate	39	0.4
PPRT (low-cost flats)	242	2.5
Squatters	46	0.5
New Village Settlement	50	0.5
Fisherman Village	6	0.1
Housing Estate	5631	57.4
Others ***	106	1.1
**Duration of Stay:**		
≤1 year	286	2.9
2–5 years	1387	14.1
6–10 years	2614	26.6
11–15 years	1612	16.4
16–20 years	1820	18.5
≥21 years	2099	21.4
**Level of Education:**		
Unfinished formal education	903	9.2
Informal education (Pondok/Madrasah)	75	0.8
Primary School	263	2.7
Secondary School	4832	49.2
University/College	3745	38.1
**Household Income Category:**		
Unemployed ****	2921	29.8
B40 (<RM 4850/USD 1155)	6514	66
M40 (RM 4850/USD 1155-RM 10,970/USD 2612)	346	3.6
T20 (>RM 10,970/USD 2612)	37	0.3

* Siam, Punjabi; ** Atheist, Free Thinker, Sikh; *** Kampung Baru, Quarters, Room Rent; **** Including students, FELDA/FELCRA (resettlement of rural poor into newly developed areas and to organize smallholder farms growing cash crops), B40, M40, and T20 (household income classifications in Malaysia: bottom 40%, middle 40%, and top 20%).

**Table 2 ijerph-19-04684-t002:** Prevalence of substance (tobacco and alcohol) and drug use among the public.

Youth Age	Lifetime					Current Use				
	Count	Estimated Population	Prevalence (%)	95% CI		Count	Estimated Population	Prevalence (%)	95% CI	
				Lower	Upper				Lower	Upper
**15–40 ^1^** **(*n* = 9818)**	2376	2,489,612	22.4	21.2	23.5	1321	1,438,296	12.9	12.0	13.9
**15–18 ^2^** **(*n* = 959)**	166	190,869	13.5	10.7	16.8	123	121,126	8.5	6.3	11.5
**19–24 ^3^** **(*n* = 2541)**	558	593,855	19.9	17.8	22.1	280	315,526	10.5	9.1	12.2
**25–30 ^4^** **(*n* = 2729)**	706	797,690	24.5	17.8	22.1	376	453,061	13.9	12.2	15.8
**31–40 ^5^** **(*n* = 3589)**	946	907,198	26.2	24.2	28.3	542	548,583	15.8	14.1	17.7

CI = confidence interval; ^1^ population size = 11,129,316; ^2^ population size = 1,417,268; ^3^ population size = 2,991,695; ^4^ population size = 3,256,320; ^5^ population size = 3,464,034.

**Table 3 ijerph-19-04684-t003:** Prevalence of lifetime substance abuse by type according to age categories.

Youth Age	Cigarette					Alcohol				
	Count	Estimated Population	Prevalence (%)	95% CI		Count	Estimated Population	Prevalence (%)	95% CI	
				Lower	Upper				Lower	Upper
**15–40 ^1^** **(*n* = 9818)**	2200	2,217,279	19.9	18.9	21.0	427	549,785	4.9	4.2	5.7
**15–18 ^2^** **(*n* = 959)**	152	159,674	11.3	9.2	13.8	19	41,771	2.9	1.3	6.3
**19–24 ^3^** **(*n* = 2541)**	513	529,749	17.7	15.8	19.8	123	130,829	4.4	3.4	5.6
**25–30 ^4^** **(*n* = 2729)**	667	736,369	22.6	20.6	24.8	135	165,441	5.1	3.9	6.6
**31–40 ^5^** **(*n* = 3589)**	868	791,487	22.8	21.1	24.7	150	211,744	6.1	4.7	7.9

CI = confidence interval; ^1^ population size = 11,129,316; ^2^ population size = 1,417,268; ^3^ population size = 2,991,695; ^4^ population size = 3,256,320; ^5^ population size = 3,464,034.

**Table 4 ijerph-19-04684-t004:** Prevalence of current substance abuse by type according to age categories.

Youth Age	Cigarette					Alcohol				
	Count	Estimated Population	Prevalence (%)	95% CI		Count	Estimated Population	Prevalence (%)	95% CI	
				Lower	Upper				Lower	Upper
**15–40 ^1^** **(*n* = 9818)**	1239	1,280,126	11.5	10.7	12.3	313	389,545	3.5	2.9	4.2
**15–18 ^2^** **(*n* = 959)**	111	94,202	6.6	5.2	8.4	15	33,876	2.4	0.9	6.0
**19–24 ^3^** **(*n* = 2541)**	260	281,698	9.4	8.0	11.0	90	84,511	2.8	2.1	3.8
**25–30 ^4^** **(*n* = 2729)**	363	428,413	13.2	11.5	15.0	103	116,982	3.6	2.6	4.9
**31–40 ^5^** **(*n* = 3589)**	505	475,813	13.7	12.3	15.3	106	154,175	4.5	3.2	6.1

CI = confidence interval; ^1^ population size = 11,129,316; ^2^ population size = 1,417,268; ^3^ population size = 2,991,695; ^4^ population size = 3,256,320; ^5^ population size = 3,464,034.

**Table 5 ijerph-19-04684-t005:** Prevalence of lifetime and current drug use by age group.

Youth Age	Lifetime					Current Use				
	Count	Estimated Population	Prevalence (%)	95% CI		Count	Estimated Population	Prevalence (%)	95% CI	
				Lower	Upper				Lower	Upper
15–40 ^1^ (*n* = 9818)	661	616,874	5.5	5.0	6.2	431	394,133	3.5	3.1	4.1
15–18 ^2^ (*n* = 959)	20	35,422	2.5	1.0	5.9	19	34,297	2.4	1.0	5.8
19–24 ^3^ (*n* = 2541)	170	147,028	4.9	4.0	6.0	119	101,440	3.4	2.6	4.4
25–30 ^4^ (*n* = 2729)	252	250,839	7.7	6.4	9.2	176	157,654	4.8	3.9	6.0
31–40 ^5^ (*n* = 3589)	219	183,586	5.3	4.5	6.3	117	100,742	2.9	2.3	3.7

CI = confidence interval; ^1^ population size = 11,129,316; ^2^ population size = 1,417,268; ^3^ population size = 2,991,695; ^4^ population size = 3,256,320; ^5^ population size = 3,464,034.

**Table 6 ijerph-19-04684-t006:** Comparison of prevalence rates by severity according to age group (*n* = 9818).

Youth Age	Mild					Moderate					Severe				
	Count	Estimated Population	Prevalence (%)	95% CI		Count	Estimated Population	Prevalence (%)	95% CI		Count	Estimated Population	Prevalence (%)	95% CI	
				Lower	Upper				Lower	Upper				Lower	Upper
15–40 ^1^ (*n* = 9818)	40	37,367	0.3	0.2	0.5	245	246,429	2.2	1.8	2.7	146	110,337	1.0	0.8	1.2
15–18 ^2^ (*n* = 959)	6	6744	0.5	0.2	1.2	8	23,042	1.6	0.4	5.8	5	4512	0.3	0.1	0.9
19–24 ^3^ (*n* = 2541)	11	12,022	0.4	0.2	0.8	62	49,384	1.7	1.2	2.3	46	40,035	1.3	0.8	2.1
25–30 ^4^ (*n* = 2729)	7	6647	0.2	0.1	0.5	103	110,825	3.4	2.6	4.4	66	40,181	1.2	0.9	1.7
31–40 ^5^ (*n* = 3589)	16	11,954	0.3	0.2	0.6	72	63,178	1.8	1.4	2.5	29	25,610	0.7	0.5	1.2

CI = confidence interval; ^1^ Population size = 11,129,316; ^2^ population size = 1,417,268; ^3^ population size = 2,991,695; ^4^ population size = 3,256,320; ^5^ population size = 3,464,034.

**Table 7 ijerph-19-04684-t007:** Prevalence rate of drug use according to socio-demographic characteristics (*n* = 9818).

Duration	Lifetime						Current Use			
	No. of Cases	Estimated Population	Prevalence (%)	Confidence Interval		No. of Cases	Estimated Population	Prevalence (%)	Confidence Interval	
Socio-Demographic (Population)				Lower (%)	Upper (%)				Lower (%)	Upper (%)
**Gender:**										
Male (5,658,778)	631	573,777	10.1	9.1	11.3	416	373,556	6.6	5.7	7.6
Female (5,470,539)	30	43,098	0.8	0.4	1.4	15	20,577	0.4	0.2	0.7
**Marital Status:**										
Single (6,311,707)	391	387,917	6.1	5.3	7.2	288	270,263	4.3	3.6	5.1
Married (4,416,318)	234	203,153	4.6	3.9	4.1	119	106,810	2.4	1.9	3.0
Divorced (401,290)	36	25,805	6.4	5.4	9.8	24	17,061	4.3	2.4	7.5
**Level of Education:**										
Unfinished formal education (730,133)	69	68,390	9.4	5.7	15.0	58	57,878	7.9	4.5	13.6
Informal education (Pondok/Madrasah) (73,124)	1	479	0.7	0.1	4.5	1	479	0.7	0.1	4.5
Primary School (288,467)	14	12,563	4.4	2.4	7.7	13	12,163	4.2	2.3	7.6
Secondary School (5,521,284)	387	340,858	6.2	5.4	7.1	270	237,433	4.3	3.7	5.0
University/College (4,516,309)	44	194,584	4.3	3.5	5.2	89	86,180	1.9	1.4	2.6
**Household Income** **Category:**										
Unemployed * (3,609,472)	78	54,116	1.5	1.1	2.0	51	31,601	0.9	0.6	1.3
B40 (7,016,411)	576	553,031	7.9	7.0	8.9	376	359,353	5.1	4.4	6.0
M40 (464,216)	6	4490	1.0	0.4	2.0	4	3179	0.7	0.2	2.1
T20 (39,217)	1	5238	13.4	2.5	53.4	-	-	-	-	-

* Including students. B40, M40, and T20 (household income classifications in Malaysia): bottom 40%, middle 40%, and top 20%.

**Table 8 ijerph-19-04684-t008:** Prevalence of drug use by drug type (*n* = 9818).

Type of Drug	Lifetime					Current Use				
	No. of Cases	Estimated Population	Prevalence (%)	Confidence Interval		No. of Cases	Estimated Population	Prevalence (%)	Confidence Interval	
				Lower (%)	Upper (%)				Lower (%)	Upper (%)
^1^ Cannabis	287	279,128	2.5	2.1	3.0	219	210,443	1.9	1.5	2.4
^2^ Cocaine	157	109,502	1.0	0.7	1.4	137	91,899	0.8	0.6	1.2
^3^ ATS	209	219,391	2.0	1.6	2.5	188	188,442	1.7	1.3	2.1
^4^ Solvents/Inhalants	86	76,474	0.7	0.4	1.1	77	65,638	0.6	0.4	1.0
^5^ Sedatives or sleeping pills	88	82,532	0.7	0.5	1.2	76	67,503	0.6	0.4	1.0
^6^ Hallucinogens	92	84,324	0.8	0.5	1.2	82	65,638	0.6	0.4	1.0
^7^ Opiates	91	80,454	0.7	0.5	1.1	77	65,382	0.6	0.3	1.0
^8^ Others	333	311,525	2.8	2.4	3.3	181	170,699	1.5	1.2	2.0

CI = Confidence interval; Population size = 11,129,316. ^1^ Cannabis, hashish or hash, marijuana, and others. ^2^ Coke, crack, and others. ^3^ Amphetamine-type stimulants (ATS): methamphetamine, ice, ecstasy, stones, Ya ba pills, and others. ^4^ Nitrus, glue, petrol, thin paint, etc. ^5^ Diazepam, alprazolam, midazolam, and others. ^6^ Lysergic acid diethylamide (LSD), acids, mushrooms, trips, ketamine, etc. ^7^ Heroin, morphine, methadone, buprenorphine, codeine, and others. ^8^ Kratom, depressants, and dissociatives (ketamine, dextromethorphan, nitrous oxide, phencyclidine, and salvia divinorum).

## Data Availability

The data presented in this study are available within the article.

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
