# Peer review of "Prevalence of Drug and Substance Use among Malaysian Youth: A Nationwide Survey"

_ijerph, 2022, doi:10.3390/ijerph19084684_

Round 1

Reviewer 1 Report

The study is well designed and the methods and results are clearly described. I would like to suggest the authors reorganizing the discussion to enable the reader to see the results in a wider context. For example:

  1. The results as compared to the previously available data
  2. Data on DSA from other countries in the region
  3. Worldwide data
  4. Interpretation- why the rate increased what might be the cultural, political etc. reason of the trend . It is particularly interesting speaking about Muslim country. It would be also interesting to see the covid19 context in most European countries DSA increased due to pandemic.

I find the paper important from the public health perspective, but as it is being addressed to the international scientific community more interpretation should be added to the data acquired in the study.

Author Response

Reviewer 1

The study is well designed and the methods and results are clearly described. I would like to suggest the authors reorganizing the discussion to enable the reader to see the results in a wider context. For example:

1. The results as compared to the previously available data

Thank you for your comment. Results were compared to Adolescent Health Survey 2017 and National Health and Morbidity Survey 2019 (line 360 to 362)

2. Data on DSU from other countries in the region

Thank you for your comment. Results were compared (line 365 to 367)

3. Worldwide data

Thank you for your comment. Results were compared (line 365 and 367)

4. Interpretation- why the rate increased what might be the cultural, political etc. reason of the trend. It is particularly interesting speaking about Muslim country. It would be also interesting to see the covid19 context in most European countries DSA increased due to pandemic.

Thank you for your comment. We had explained the discrepancy of prevalence rate between local and abroad studies “These differences in findings may be due to this study’s method of data gathering (i.e., face-to-face interviews), sampling size, and respondent characteristics (youth aged 15-40 years), since other studies only used high school and university students as their samples. Other factors may be due to differences in culture and social ceremonies. In addition, inherited ritual ceremonies may sometimes lead to substance intake that corresponds to social norms or to beliefs such as those reflected in orally transmitted traditions related to the substance types. Conversely, drug and alcohol consumption may also be incompatible with local norms and culture, especially for Muslims. Furthermore, the respondents’ geographical location may also have affected the findings, especially for those who live near the country’s international border entrances” (line 367 to 376).

I find the paper important from the public health perspective, but as it is being addressed to the international scientific community more interpretation should be added to the data acquired in the study.

Thank you for your comment. We have added more interpretations to the data in the table footnotes (line 200 to 201; 336; 344; 347; 349)

Thank you.

Reviewer 2 Report

This article aims to determine the prevalence of drug and substance use among Malaysian youths. The work is interesting and well-conducted, even if so much restricted to the Malaysian case. Nonetheless, the English is very poor and typos and grammatical errors are spread all over the article. All the following comments should be considered as minor suggestions, but a complete revision of the language has to be taken for given from so on, and is strongly advised.

  • The Introduction is accurate, exhaustive, and well-documented.
  • The sampling strategy is very accurate, convincing and well-explained. The sample size is convincing. It is really rare to find a study with such a randomization, nowadays. Good job.
  • Section 2.3 is somehow unclear, at least for me. What is an enumerator and how it relates with COVID measures? What are validated instruments in this framework? Can you be a little more specific about how ASSIST scale works?

Regarding the analysis, everything seems fine. Here, just some minor suggestions for improving the quality and the appeal of the overall work:

  • Please, in the results’ exposition try to stress more not (or not only) the age group with highest prevalence, but the youngest age group, since in the title you state the paper is focused on youths. Other age groups are useful for comparisons, anyway.
  • You should consider to compute also coefficients of variation: indeed, as much an individual is young, the less time she has to start using substances.
  • Are you able to integrate your conclusions with: (i) an evaluation of the new rates of prevalence calculated from the social effects point of view in Malaysia (is it too high or in line with expectations?); (ii) a comparison with data from other countries, maybe also bigger or richer than Malaysia (the prevalence is higher or lower than other countries’ one? Could it be due to availability of substances or to the country’s GDP or other factors?)

Author Response

Reviewer 2

This article aims to determine the prevalence of drug and substance use among Malaysian youths. The work is interesting and well-conducted, even if so much restricted to the Malaysian case. Nonetheless, the English is very poor and typos and grammatical errors are spread all over the article. All the following comments should be considered as minor suggestions, but a complete revision of the language has to be taken for given from so on, and is strongly advised.

  • The Introduction is accurate, exhaustive, and well-documented.

Thank you.

  • The sampling strategy is very accurate, convincing and well-explained. The sample size is convincing. It is really rare to find a study with such a randomization, nowadays. Good job.

Thank you.

  • Section 2.3 is somehow unclear, at least for me. What is an enumerator and how it relates with COVID measures? What are validated instruments in this framework? Can you be a little more specific about how ASSIST scale works?

Thank you for your comment. Enumerator is person employed in taking a census of the population. We employed few enumerators from each state to collect data from the selected individuals in the living quarters. They were briefed on COVID-19 fieldwork safety protocol such as wearing mask and maintaining social distancing while interviewing.

Thank you for your comment. This study used various validated instruments such as Depression, Anxiety and Stress Scale (DASS 21), Religious Commitment Inventory (RCI-10), Self-Esteem Rosemberg Scale, Resilience Scale, and many more. However, we reported ASSIST questionnaire only in this article as it is only describe the prevalence of drug and substance abuse.

Thank you for your comment. We have added the information on ASSIST in the method section (subtitle 2.3 Data Collection and Research Instruments) “ASSIST determines a risk score for each substance by placing it in a lower-risk, moderate-risk, or high-risk category. It obtains information from respondents about their lifetime use of substances, as well as problems associated with substance use (acute intoxication, regular use, dependent and injecting behavior) over the last three months (current users)” (lines 149 to 153).

Regarding the analysis, everything seems fine. Here, just some minor suggestions for improving the quality and the appeal of the overall work:

  • Please, in the results’ exposition try to stress more not (or not only) the age group with highest prevalence, but the youngest age group, since in the title you state the paper is focused on youths. Other age groups are useful for comparisons, anyway.

Thank you for your suggestion. We have added the statement of “More concerning is the prevalence of current tobacco and alcohol use among those of secondary school age (15 to 18 years old) which was found to be 6.6% and 2.4%, respectively.” (line 214 to 216).

  • You should consider to compute also coefficients of variation: indeed, as much an individual is young, the less time she has to start using substances.

Thank you for your suggestion and comment. We would like to keep our presentation in 95% confidence interval as this measures how well our samples represent the population that were studied. The coefficient of variation is recommended for data measured on scales that have a meaningful zero. We hope you can consider this explanation. 

  • Are you able to integrate your conclusions with:
  • (i) an evaluation of the new rates of prevalence calculated from the social effects point of view in Malaysia (is it too high or in line with expectations?);

Thank you for your comment. We have added the information in conclusion segment “…. is higher than that previously reported among adolescents and adults in Malaysia.” (line 433 to 434).

  • (ii) a comparison with data from other countries, maybe also bigger or richer than Malaysia (the prevalence is higher or lower than other countries’ one? Could it be due to availability of substances or to the country’s GDP or other factors?)

Thank you for your comment. We have added the explanations in the discussion segment “… this study generally found lower prevalence rates compared to studies from abroad, such as United States, Norway, and Iran. These differences in findings may be due to this study’s method of data gathering (i.e., face-to-face interviews), sampling size, and respondent characteristics (youth aged 15-40 years), since other studies only used high school and university students as their samples. Other factors may be due to differences in culture and social ceremonies. In addition, inherited ritual ceremonies may sometimes lead to substance intake that corresponds to social norms or to beliefs such as those reflected in orally transmitted traditions related to the substance types. Conversely, drug and alcohol consumption may also be incompatible with local norms and culture, especially for Muslims. Furthermore, the respondents’ geographical location may also have affected the findings, especially for those who live near the country’s international border entrances” (line 365 to 376).

Thank you.

Reviewer 3 Report

Thank you for submitting your manuscript “Prevalence of Drug and Substance Use Among Malaysian Youths: A Nationwide Survey”. It is an interesting read and I can appreciate there is underreporting of drug and substance use in multiple countries if not most countries. The methods overall are solid. I do have some queries which are detailed below. The discussion has a good framework. I would suggest defining all abbreviations on first use.

Introduction

It is unclear what “For trend wise analytics” is. Do the authors mean accurate or relevant analyses?

International readers might not know what kratom is. Could a statement be added e.g. a herbal extract that works like opioids, or whatever is the most suitable description.

Why are adolescents more prone to addiction? (line 76). Is the answer line 77-80? I was expecting the answer to be in the next sentence.

So, is this study including “drug addicts”? (last paragraph of the introduction). I was expecting the last sentence to confirm this but it does not. Otherwise, I am unsure why this paragraph is needed relative to this study. I acknowledge what the authors are stating but its relevance is unclear.

Methods

How was the upper limit age of youths determined to be 40 years? The WHO and United Nations definition of youth is people 15-24 years.

What was the definition of “non-Malaysian”? Was it someone not born in Malaysia? Or was it related to heritage?

Is there literature to support a 20% non-response rate? How was this value chosen?

Regarding the p-value of 0.305 on line 128, is this correct?

Line 135 refers to “informants”, do the authors mean youth participants?

How many enumerators were there?

Were participants reimbursed? How truthful were participants in their responses?

One in a lifetime and lifetime prevalence of drug use are very different. What was the rationale to group these severity of drug use together? Possibly my definition of lifetime prevalence is different to the authors. To avoid confusion for readers, these terms need defining.

Was there any missing data? How was it handled?

Was data collected electronically or on paper? how was data managed (e.g. data entry, integrity checking etc).

Results

In section 3.1, 64.9% were Malays. I thought non-Malays were excluded? (line 118).

Table 1

Background information on household income is needed for international readers.

Also, there are undefined abbreviations e.g. in “types of settlers”. If the details are not in the methods, then a footnote is needed.

Table 2-5: It is strange to report 95%CI per row (i.e. one row for lower and one row for upper CI) and not adjacent to the point estimate.

Abbreviations need to be defined e.g. Table 5. What is B40, M20, T20?

Table 7. Typo “7” prefix. Was oxycodone reported? What is an example of “Dissociative”? What is ATS?

Discussion

I would consider splitting section 4.1 and separating “Core Summary Findings” (i.e. one paragraph) and “Comparisons with Existing Literature”.

References

References #15, 48 are incomplete.

Author Response

Reviewer 3

Thank you for submitting your manuscript “Prevalence of Drug and Substance Use Among Malaysian Youths: A Nationwide Survey”. It is an interesting read and I can appreciate there is underreporting of drug and substance use in multiple countries if not most countries. The methods overall are solid. I do have some queries which are detailed below. The discussion has a good framework. I would suggest defining all abbreviations on first use.

Thank you for your comment. Abbreviations have been defined (line 139; 200 to 202; 279; 336; 344; 347; 349; 403).

Introduction

It is unclear what “For trend wise analytics” is. Do the authors mean accurate or relevant analyses?

Thank you for your comment. We meant relevant analyses as the statement came from a large population based epidemiological study in Sweden.

International readers might not know what kratom is. Could a statement be added e.g. a herbal extract that works like opioids, or whatever is the most suitable description.

Thank you for your comment. We have added the statement to describe kratom as “a herbal substance that can produce opioid- and stimulant-like effects” (line 65 and 66).

Why are adolescents more prone to addiction? (line 76). Is the answer line 77-80? I was expecting the answer to be in the next sentence.

Thank you for your comment. We have rearranged the explanation of adolescents who are more prone to addiction soon after the statement (line 79 to 81).

So, is this study including “drug addicts”? (last paragraph of the introduction). I was expecting the last sentence to confirm this but it does not. Otherwise, I am unsure why this paragraph is needed relative to this study. I acknowledge what the authors are stating but its relevance is unclear.

Thank you for your comment. This study does not include drug addicts. We have deleted the last paragraph of the introduction.

Methods

How was the upper limit age of youths determined to be 40 years? The WHO and United Nations definition of youth is people 15-24 years.

Thank you for your comment. We used definition of youths based on the characterization from Malaysian Youth Council and Ministry of Youth and Sports, Malaysia as stated in Youth Societies and Youth Development Act 2007 (Act 668). The explanation has been added in the text, “from the definition of youth based on the Malaysian Youth Council and the Ministry of Youth and Sports, Malaysia, as stated in the Youth Societies and Youth Development Act 2007” (line 110 to 112).

What was the definition of “non-Malaysian”? Was it someone not born in Malaysia? Or was it related to heritage?

Thank you for your questions. Non-Malaysian is define as an individual who does not hold a blue identity card issued by the National Registration Department of Malaysia. Not necessary born in Malaysia.

Is there literature to support a 20% non-response rate? How was this value chosen?

Thank you for the questions. We chose that non-response rate based on recommendation by Jack E. Fincham (PhD) in his article titled ‘Response Rates and Responsiveness for Surveys, Standards, and the Journal. American Journal of Pharmaceutical Education 2008; 72(2) Article 43. This reference has been added in the text (line 123) and in the reference list (line 510 and 511).

Regarding the p-value of 0.305 on line 128, is this correct?

Thank you for the comment. We admit the typo error. The p value is now deleted.

Line 135 refers to “informants”, do the authors mean youth participants?

Thank you for your comment. Yes, we referred the ‘informants’ as the participants. We have changed the word in the text.

How many enumerators were there? 

There were around 15 to 20 enumerators assigned for each state and they were supervised by two field managers. We have added “Trained enumerators administered validated instruments during the house-to-house data collection and were supervised by field managers” in the text (line 140 to 141).

Were participants reimbursed? How truthful were participants in their responses?

Thank you for your comment. We did not reimburse the respondents for their participation. It is solely voluntary. We assumed truthfulness of the participants’ responses as no data on the name of the respondent captured.

One in a lifetime and lifetime prevalence of drug use are very different. What was the rationale to group these severity of drug use together? Possibly my definition of lifetime prevalence is different to the authors. To avoid confusion for readers, these terms need defining.

Thank you for your comment. We have standardized the usage of lifetime prevalence and we have defined it in the text (line 160 to 161). We have added “users who at some point in their lives have used the drug before the time of assessment”.

Was there any missing data? How was it handled?

Thank you for the questions. All the data were collected electronically using software application. Each of the answer for the questions must be entered before the other segments can be continued. We have added the statement “The data were collected electronically by the enumerators using a software application. The verification of the data was conducted by field managers and the researchers” (line 155 to 157).  

Was data collected electronically or on paper? how was data managed (e.g. data entry, integrity checking etc).

Data were collected electronically. The data were entered by the enumerators, verified by the field managers and lastly by the researchers. We have added the statement “The data were collected electronically by the enumerators using a software application. The verification of the data was conducted by field managers and the researchers” (line 155 to 157).  

Results

In section 3.1, 64.9% were Malays. I thought non-Malays were excluded? (line 118).

Thank you for the comment. Malay is one of the ethnicity in Malaysia. The other main ethnicities are Chinese and Indian. We excluded non-Malaysian and this refers to nationality.

Table 1

Background information on household income is needed for international readers.

Thank you for your comment. The explanation of household income were added, “B40, M40, and T20 (household income classifications in Malaysia: bottom 40%, middle 40%, and top 20%” (line 201 to 202).

Also, there are undefined abbreviations e.g. in “types of settlers”. If the details are not in the methods, then a footnote is needed.

Thank you for your comment. The explanation of types of settlers were added “resettlement of rural poor into newly developed areas and to organize smallholder farms growing cash crops” (line 200 to 201).

Table 2-5: It is strange to report 95%CI per row (i.e. one row for lower and one row for upper CI) and not adjacent to the point estimate.

Thank you for your comment. Rearrangement of the confidence intervals have been made (Table 2, 3, 4 and 5).

Abbreviations need to be defined e.g. Table 5. What is B40, M20, T20?

Thank you for your comment. The explanation of household income were added “B40, M40, and T20 (household income classifications in Malaysia: bottom 40%, middle 40%, and top 20%” (line 201 to 202).

Table 7. Typo “7” prefix. Was oxycodone reported? What is an example of “Dissociative”? What is ATS?

Thank you for your questions. Typo ‘7’ error has been corrected. Oxycodone is a type of opioids. We did captured opioids in this study but not specifically on oxycodone.

We have added the examples of dissociative as “ketamine, dextromethorphan, nitrous oxide, phencyclidine and salvia divinorum” (line 349).

We have added the explanation of ATS as “Amphetamine Type Stimulants” (line 344).

Discussion

I would consider splitting section 4.1 and separating “Core Summary Findings” (i.e. one paragraph) and “Comparisons with Existing Literature”.

Thank you for your suggestion. However, our discussion embedded the findings and the comparison with literature reviews. It is quite difficult for us to separate it into different sub sections. We hope that our justification would be acceptable to you.

References

References #15, 48 are incomplete.

Thank you for the comment. We have added the information on the references no 15 (line 486 to 487) and 48 (now become no 50) (line 554 to 556).

Thank you.